

A Gauss Elimination Method for estimating locations of extrema in gridded data:
Applications for Potential Field Data
Dung Nguyen Kim[a,*], Dung Tran Tuan[a,b]
[a] Institute of marine geology and geophysics, Vietnam Academy of Science and Technology.
[b] Graduate University of Science and Technology, Vietnam Academy of Science and Technology
18 Hoang Quoc Viet Street, Cau Giay District, Hanoi City, Vietnam
[*] kimdunggeo@yahoo.com
**Abstract:**
Extrema in gravity measurements can be used to locate geological structures of
interest and the boundaries of such structures can be associated with the maxima in the
gradients of the gravitational field strength. Finding the extrema of measured geophysical
fields measured on the Earth's surface when the data is sparse is challenging. The inferred
positions of such extrema are highly model dependent. Polynomial functions of two variables
can be fitted to the data. Higher order polynomials typically give more accurate determination
of the extrema, but the maximum order of the polynomial is limited by the number of data
points. Difficulties are accentuated in the vicinity of boundaries of the existing data. The
maximum horizontal gradient method has often been applied in this context. But in that
particular construction, quadratic functions are developed in each dimension. Although the
magnitudes of the extracted coefficients are obtained from three points related by their
positions on orthogonal straight lines, off axis information should be included as well. The
present paper introduces a modification of the maximum horizontal gradient method to
overcome these difficulties. A Function f of the two variables x and y:
$f_{(x,y)} = a_1 x^2 + a_2 y^2 + a_3 x^2 y^2 + a_4 x^2 y + a_5 x y^2 + a_6 xy + a_7 x + a_8 y + a_9$ is    established    by
Gaussian elimination method base on a 3x3 neighborhood data grid. An extract creates a 4-
dimensional space based on 4 specific cases of function f, including x = 0, y = 0, y = -x and y
= x, they are four functions of one variable. The extreme points position are detected from
these functions of one variable. To prove the proposed theoretical basis, as well as the built
computer program, the paper presents two numerical models. The obtained results shown that
the new approach has more maxima points than the traditional approach. Beside advantages
of new approach, some disadvantages is also discussed in this paper. Moreover, we conclude
with the application of our new approach to gravitational data in the East Vietnam Sea and
demonstrate that we thereby disclose the existence of a gravity trench undetectable in the
traditional method.


## 1. Introduction


We have many the methods, as well as the approach, can estimate geological
boundaries. These methods studied very detailed from the theoretical basis to the numerical
models and applied for the real data. In these methods, firstly, we mentioned to the
normalized full gradient of gravity anomaly method (NFG) of Berezkin, W.M, 1967 [5].
After that, it is applied, improved and developed by the scientists such as:
Karsli.H.,Bayrak.Y., 2010[16], Oruc B.,2008, 2012[18,19], Ebrahimzadeh Ardestani. V,
2004[10], Ekinci, Y.L., [11,13], Aghajani.H, 2009[1], Sheng.Z,2015 [25]. The horizontal
gradient method of Cordell, 1979 [8]. The maximum horizontal gradient method of Blakely,
R, J., Simpson, R.W, 1986 [6] and Cordell. L, Grauch. V. J. S, 1985[7]. Using two-variables
function to detect the extreme points of Phillips, J.D. 2007[24],....Or the methods use the
components of gradient tensor (analytic signal, directional derivatives,...) to approximate
geological boundaries and estimate depth simultaneously, such as: Beiki M, (2010,2011
[2,3,4]), Ekinci, Y.L., [11,12, 13], Pedersen L.B, 1990 [23], Oruc, B., 2012, 2013 [20,21],
Kim Dung N., 2016 [17],... Each author, as well as each method, has the different approach,
but the common goal is detect geological boundaries and increase the accuracy of method.
These methods are very powerful. They confirmed on many papers and projects of authors,
applied for the geological structure research, oil and gas exploration and exploitation, mineral
resources on the world.
However, each method has advantages and disadvantages. In this paper, we only
discuss about the method of Blackely, R. J., Simpson, R.W, 1986 [6] and the method of
Phillips, J.D. 2007 [24]. In the method of Blackely,R.J, the limit of method can only detect
the maxima point that lies on four dimensions and is the maximum point of a quadratic
function that is established by three points on a straight line. Thus, the accuracy of the
approximated geological boundaries aren't high enough. In the method of Phillips, J.D,
Function of two variables is established base on a 3x3 neighborhood data grid and used to
detect the maximum point. Therefore, the maximum point isn't only lies on four dimensions
such as the method of Blackely, R. J.,but also can lies anywhere within 3x3 grid points, it is
advantage of this method. Nevertheless, the coefficients determination for quadratic surface
of Phillips, J.D have unstable accuracy because the equation number is nine whereas the
variable number (the coefficient number) is six, thus, it isn't unique root. For these reasons,
originating from two-variables function, the paper proposes a new approach, a algorithm that
use Gauss elimination method to determine the coefficients of two-variables function base on
a 3x3 neighborhood data grid, it holds the geophysics characteristic. Using Gauss elimination



method is the marked differences between our approach and Phillips, J.D's approach. After
two-variables function is established, the paper examine four specific cases, including x=0,
y=0, y=-x and y=x, they correspond with four functions of one variable. These functions are
very different from the functions that the proposed Blakely, R, J.,. The maxima points are
detected from these functions.

**2. Method**
The paper researches a function of two variables that has pattern:
$f_{(x,y)} = a_1 x^2 + a_2 y^2 + a_3 x^2 y^2 + a_4 x^2 y + a_5 x y^2 + a_6 xy + a_7 x + a_8 y + a_9;$   (1)
To determine coefficients $a_i$ of function $f_{(x,y)}$ (1), we use Gauss elimination method.
Namely, from function $f_{(x,y)}$, we can write 9 equations that correspond with 9 data points (3x3
data grid), including: $ij^{th}$ point and 8 neighborhood points (F*ig.1*):
- 1$^{st}$ point: $a_1 x_1^2 + a_2 y_1^2 + a_3 x_1^2 y_1^2 + a_4 x_1^2 y_1 + a_5 x_1 y_1^2 + a_6 x_1 y_1 + a_7 x_1 + a_8 y_1 + a_9 = g_1;$ (2)

............................................................................................................................

- 9$^{th}$ point: $a_1 x_9^2 + a_2 y_9^2 + a_3 x_9^2 y_9^2 + a_4 x_9^2 y_9 + a_5 x_9 y_9^2 + a_6 x_9 y_9 + a_7 x_9 + a_8 y_9 + a_9 = g_9;$ (3)
In which, $x_1 \div x_9$ and $y_1 \div y_9$ are co-ordinate (x,y) of data points from 1 to 9. For these
equations, we can build the supplemental matrix in the form:
$$A_{bs} = \begin{pmatrix} x_1^2 & y_1^2 & x_1^2 y_1^2 & x_1^2 y_1 & x_1 y_1^2 & x_1 y_1 & x_1 & y_1 & 1 \\ x_2^2 & y_2^2 & x_2^2 y_2^2 & x_2^2 y_2 & x_2 y_2^2 & x_2 y_2 & x_2 & y_2 & 1 \\ x_3^2 & y_3^2 & x_3^2 y_3^2 & x_3^2 y_3 & x_3 y_3^2 & x_3 y_3 & x_3 & y_3 & 1 \\ x_4^2 & y_4^2 & x_4^2 y_4^2 & x_4^2 y_4 & x_4 y_4^2 & x_4 y_4 & x_4 & y_4 & 1 \\ x_5^2 & y_5^2 & x_5^2 y_5^2 & x_5^2 y_5 & x_5 y_5^2 & x_5 y_5 & x_5 & y_5 & 1 \\ x_6^2 & y_6^2 & x_6^2 y_6^2 & x_6^2 y_6 & x_6 y_6^2 & x_6 y_6 & x_6 & y_6 & 1 \\ x_7^2 & y_7^2 & x_7^2 y_7^2 & x_7^2 y_7 & x_7 y_7^2 & x_7 y_7 & x_7 & y_7 & 1 \\ x_8^2 & y_8^2 & x_8^2 y_8^2 & x_8^2 y_8 & x_8 y_8^2 & x_8 y_8 & x_8 & y_8 & 1 \\ x_9^2 & y_9^2 & x_9^2 y_9^2 & x_9^2 y_9 & x_9 y_9^2 & x_9 y_9 & x_9 & y_9 & 1 \end{pmatrix} \begin{vmatrix} g_1 \\ g_2 \\ g_3 \\ g_4 \\ g_5 \\ g_6 \\ g_7 \\ g_8 \\ g_9 \end{vmatrix}$$
  (4)

If we use local coordinate system and consider $g_{(i,j)}$ point (5$^{th}$ point on F*ig.1*) is
coordinate origin (coordinate (0,0)) and the distance between two data points on datum line
ox is Δx, on datum line oy is Δy. Put them into matrix (4) and using Gauss elimination
method, we will obtain a triangle matrix:


$$A_{bs} = \begin{pmatrix} \Delta x^2 & \Delta y^2 & \Delta x^2 \Delta y^2 & \Delta x^2 \Delta y & -\Delta x\,\Delta y^2 & -\Delta x\,\Delta y & -\Delta x & \Delta y & 1 \\ 0 & \Delta y^2 & 0 & 0 & 0 & 0 & 0 & \Delta y & 1 \\ 0 & 0 & -\Delta x^2 \Delta y^2 & -\Delta x^2 \Delta y & \Delta x\,\Delta y^2 & \Delta x\,\Delta y & 2\Delta x & 0 & 1 \\ 0 & 0 & 0 & -2\Delta x^2 \Delta y & 2\Delta x\,\Delta y^2 & 0 & 2\Delta x & -2\Delta y & 0 \\ 0 & 0 & 0 & 0 & -2\Delta x\,\Delta y^2 & 2\Delta x\,\Delta y & -2\Delta x & 0 & 0 \\ 0 & 0 & 0 & 0 & 0 & 4\Delta x\,\Delta y & 0 & 0 & 0 \\ 0 & 0 & 0 & 0 & 0 & 0 & -2\Delta x & 0 & 0 \\ 0 & 0 & 0 & 0 & 0 & 0 & 0 & -2\Delta y & 0 \\ 0 & 0 & 0 & 0 & 0 & 0 & 0 & 0 & 1 \end{pmatrix} \begin{vmatrix} g_1 \\ g_2 \\ g_6 - g_1 + g_2 \\ g_9 - g_1 \\ g_7 - g_9 \\ g_3 - g_1 + g_7 - g_9 \\ g_4 - g_6 \\ g_8 - g_2 \\ g_5 \end{vmatrix} \quad (5)$$

From triangle matrix (5), we can infer coefficients $a_i$ as follows:
$a_9 = g_5; \quad a_8 = \dfrac{g_8 - g_2}{-2\Delta y}; \quad a_7 = \dfrac{g_4 - g_6}{-2\Delta x}; \quad a_6 = \dfrac{(g_3 - g_1) + (g_7 - g_9)}{4\Delta x \Delta y};$
$a_5 = \dfrac{(g_7 - g_9) + 2(g_6 - g_4) + (g_1 - g_3)}{-4\Delta x \Delta y^2}; \; a_4 = \dfrac{(g_9 - g_3) + 2(g_2 - g_8) + (g_7 - g_1)}{-4\Delta x^2 \Delta y};$
$a_3 = \dfrac{2(g_8 + g_2) + 2(g_6 + g_4) - (g_7 + g_1) - (g_9 + g_3) - 4g_5}{-4\Delta x^2 \Delta y^2};$
$a_2 = \dfrac{g_2 - 2g_5 + g_8}{2\Delta y^2}; \qquad\qquad a_1 = \dfrac{g_4 - 2g_5 + g_6}{2\Delta x^2}; \qquad (6)$
Therefore, the function of two variables $f_{(x,y)}$ is established for 9 data points (a 3x3
data grid). If we compare this paper's approach with Blackely's approach, we have a
summary table (*table 1*).

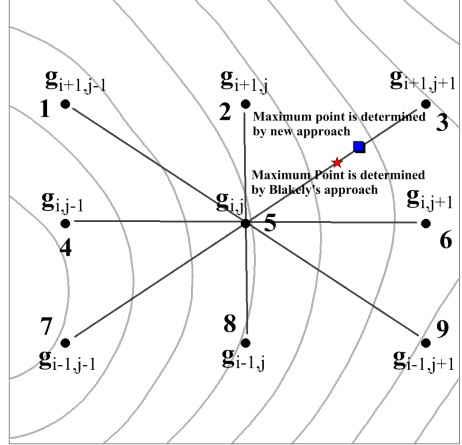

|  | Blakely R.J | This paper |
|---|---|---|
| *The number of equations* | 4 | 4 |
| *The order of polynominal* | 2 | 2 (for case: x=0, y=0); and 4 (for case: y=-x, y=x) |
| *Determine the coefficients of function* | Base on 3 points | Base on 9 points |
| *Detect the maximum point* | - $ij^{tth}$ point is larger 2 surround points (straight line) | - the extreme point is larger 2 surround points (straight line) |

Fig.1. Location of grid intersections
detect a maximum point near $g_{(i,j)}$.

Table.1. Compare Blakely's problem with
problem of this paper

To detect the maxima points of function $f_{(x,y)}$. Firstly, we have to detect the critical
points (may be maximum point, minimum point, sadle point) by the simultaneous solution of





equations $f_x = 0; f_y = 0$ . Secondly, applying the extreme conditions of two-variables
function to detect maximum point. However, this paper doesn't detect the critical points from
function $f_{(x,y)}$ but detects the extreme points from 4 functions of one variable that
corresponds with 4 specific cases of function $f_{(x,y)}$ (cases: x=0,y=0, y=-x and y=x, view
equations from A-2 to A-5 (Appendix A)). Locate of the maximum point on each dimension,
along with the corresponded condition, is determined by the expression A-8, A-9, A-12, A-13
and A-18 to A-21 (Appendix A).

Base on the proposed theoretical basis, we built a computer program by the Matlab

computer programming language to detect the extreme points on 4 dimensions.

**3. Test cases**

Hereafter, the paper use the built computer program to test on two numerical model.

The parameters of each model are given in the *table 2* and are shown on *figure 2*. In which,
the model 1 has two objects, the points number on datum-line ox and datum-line oy: nx=101,
ny=101, the distance between points on both ox and oy: dx=0.1km, dy=0.1km. The model 2
has four objects, the points number on datum-line ox and datum-line oy: nx=101, ny=101, the
distance between points on both ox and oy: dx=1km, dy=1km. The gravity anomaly, as well
as the location of objects, is created and calculated by a Matlab code that is built base on the
theoretical basis of Mantik Talwni and Maurice Ewing [26].

|  | *Model 1* | | | *Model 2* | | |
|---|---|---|---|---|---|---|
|  | Points location<br>xi/yi (km) | Depth<br>z1/z2<br>(km) | Density<br>constrast<br>(g/cm$^3$) | Points location<br>xi/yi (km) | Depth<br>z1/z2<br>(km) | Density<br>constrast<br>(g/cm$^3$) |
| Object 1 | 2.4/4 ; 2.4/6 ; 4.4/6; 4.4/4 | 0.5/2.5 | 0.1 | 24/69 ;  29/80 ;  40/85;<br>48/72 ;  34/64 ;  24/69 | 1/5 | 0.1 |
| Object 2 | 5.6/4 ; 5.6/6  ; 7.6/4 ;7.6/6 | 0.5/2.5 | 0.15 | 20/30 ; 20/40 ; 28/50 ; 36/50;<br>44/40 ; 44/30 ; 36/20 ; 28/20 | 1/6 | -0.2 |
| Object 3 |  |  |  | 60/65; 75/80; 80/75; 65/60 | 3/5 | 0.1 |
| Object 4 |  |  |  | 50/50; 50/60; 80/30; 80/20 | 1/5 | -0.2 |

*Table 2: The parameters of two models*

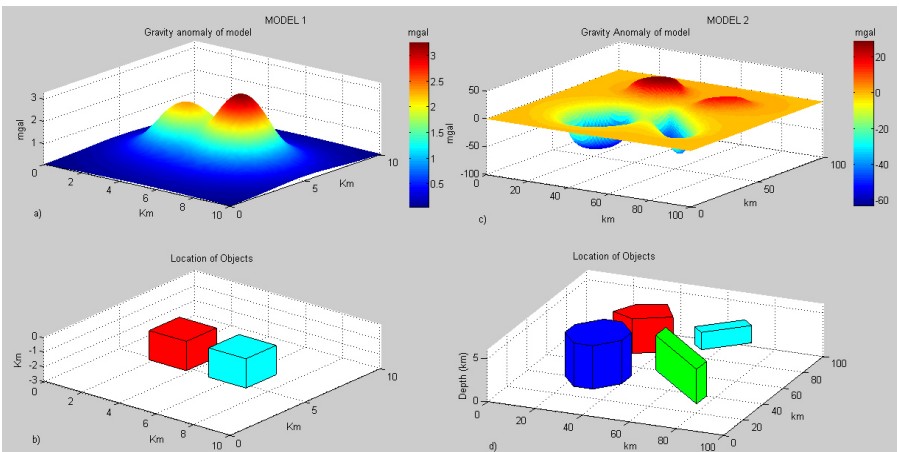


*Fig 2: a). Gravity anomaly of objects for model 1; b). Location of objects for model 1;*

*c). Gravity anomaly of objects for model 2; d). Location of objects for model 2;*

**3.1. Model 1:**
*3.1.1. Case 1: The model hasn't noise.*

Firstly, the paper test on model hasn't noise, it has only anomaly of objects (*figure 2a,*

*3a*). The maxima points are detected from the horizontal gradient amplitude function of
gravity anomaly by both approachs. The results are presented in *figure 3*. With the red points
are the result from Blakely's approach (*fig.3c*) and the blue points are the result from new
approach (*fig.3d*).

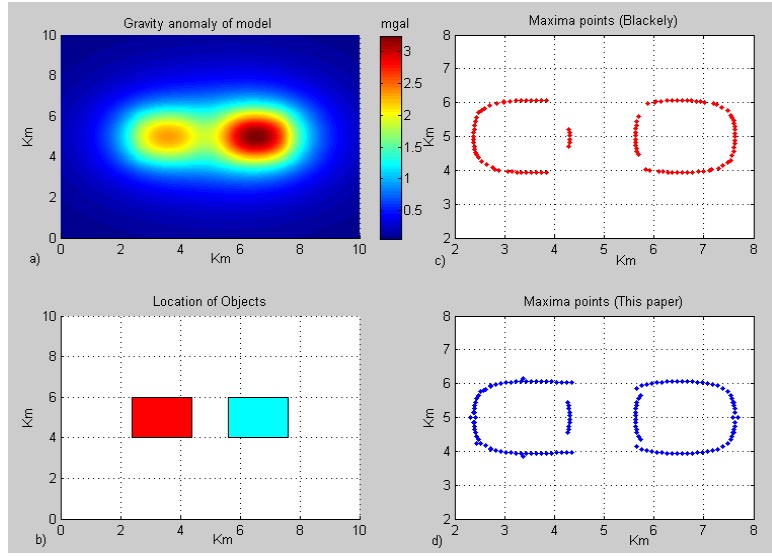


*Figure 3: The results of model hasn't noise .*

From the figure shown that the blue point is more than the red point. Namely, for
object 1, at x>4  km, the red point exists little, whereas, the blue point exists more. A
similarity for object 2,at x<6 km,we can also see the blue point is still more than the red point
*3.1.2. Case 2: The model has noise.*
To insert the noise into the model data. The author use fomular below (values from
the uniform distribution on the interval [a, b]):
Noise_Insert = a+(b-a).*rand(n,1);    with: n=length(data);
The paper test for case : a=0.08, b=0.32.
then:   Data_have_noise = model data + Noise_Insert.
A upward continuation was processed to attenuate the effect of the shortest
wavelengths:
$$Up(x,y) = F^{-1}\left\{\left(e^{-\Delta z|k|}\right)F(g)\right\};\quad \text{with}\quad \Delta z = 0.5 \text{ (km)}.$$
The maxima points are detected from the horizontal gradient amplitude function of
the upward continued field by both approachs. The results are shown on *fig. 4*. With the red
points are the result from Blakely's approach and the blue points are the result of new
approach.

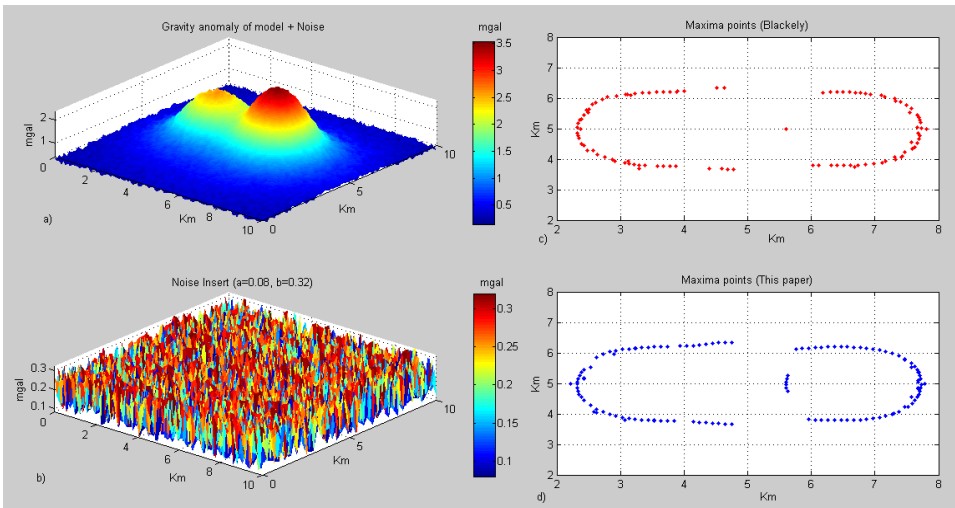

*Figure 4: The model has noise (with a=0.08, b=0.32)*
From the figure, we can see that the noise is still appear on both results (some the
maxima points appear on edge). It can be explained due to a 0.5 km upward continuation
process cann't remove the perfect noise. However, we can see the blue point is more than the
red point. Namely, for object 1, at x>4  km, the blue point is more than the red point. For





object 2, at x<6 km, the red point almost no exist (only one point) but the blue point still
appears. The boundaries of two objects can be approximated by these maxima points.
Therefore, if we use the blue points to approximate the edge of objects, it will show more
clearly in the both cases,has and hasn't noise
*Comments:* From the results obtained on this model shown:
- The proposed theoretical basis in this paper, as well as the built computer program
by this theory, is correct and logical. This is a new approach, a algorithm that can use to
detect the edge points of objects by the potential field data.
- The maxima points are detected by this algorithm are more than Blakely's approach
in the both case, hasn't noise and has noise.

### 3.2. Model 2
*3.2.1. Case 1: The model hasn't noise.*

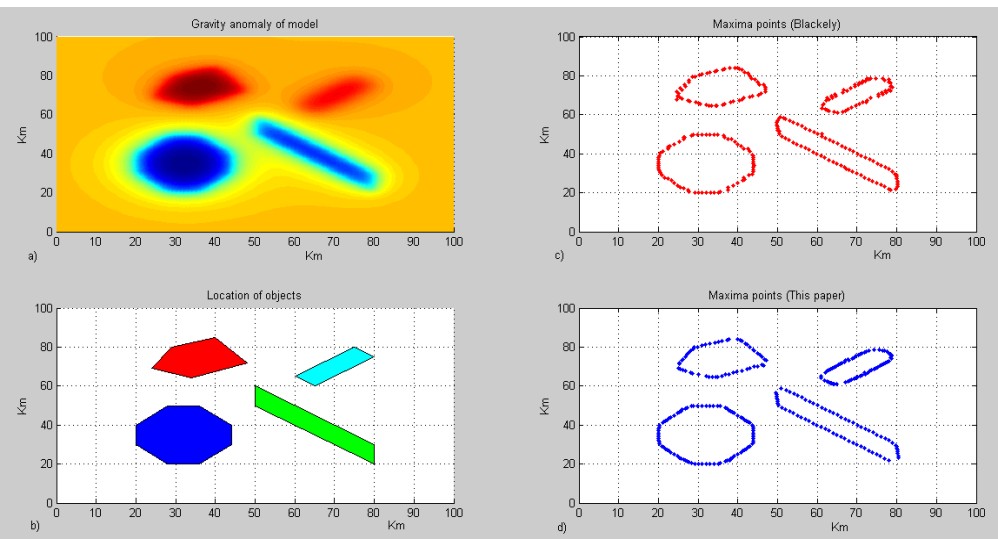

*Figure 5: The results for model 2. Case hasn't noise*
The same model 1, the paper also test two case: hasn't noise and has noise. This
model has 4 objects, they are right prisms with the top and the bottom are polygons (*fig.2d*
*and fig.5b*).
For model 1, both case, to detect the maximum point from the extreme points, the
paper can choose n>=2 (satisfying equations number). But for model 2, in this case, if we
choose n>=2, it is loosely. Therefore, the paper chose n>2 for the new approach (*figure 5d*)
and n>=2 for Blackely's approach (*figure 5c*).
*3.2.2. Case 2: The model has noise.*

The same model 1, the paper also insert noise by formular:

Noise_Insert = a+(b-a).*rand(n,1);   with a=0.08, b=0.32, n=length(data);

A 1.5km upward continuation was processed to attenuate the effect of the shortest

wavelengths. In this case, the paper chose n>2 for both the new approach and Blackely's
approach. The obtained results are shown on *figure 6c and 6d.*

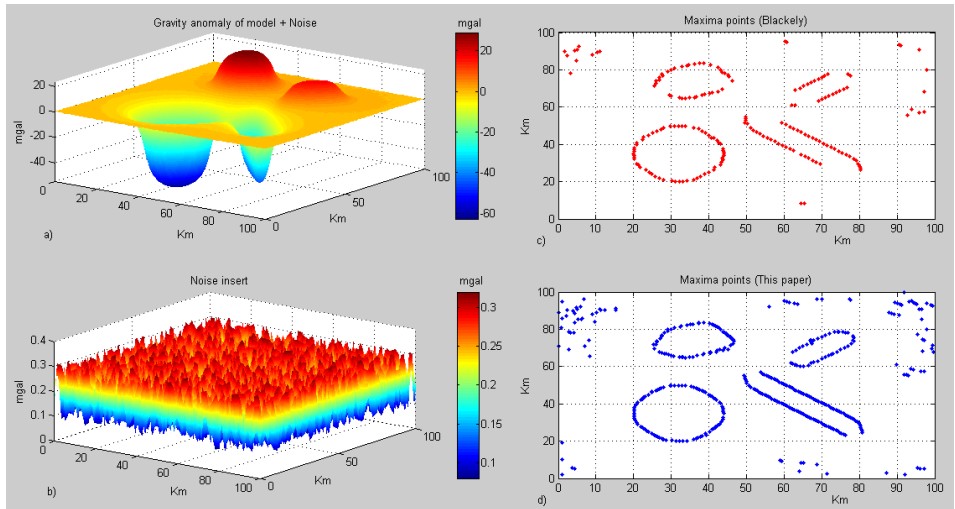

*Figure 6: The results for model 2. Case has noise*

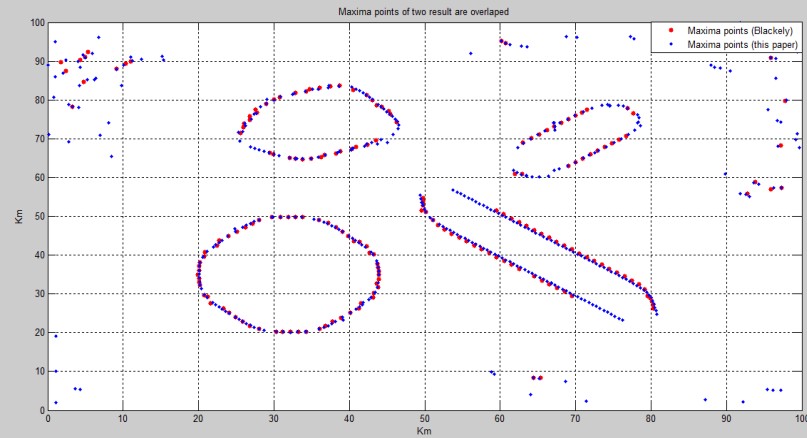

*Fig 7: The maxima points of two results are overlaped*

*\* Comments:*

*- Confirmation*: From the results are tested on model 2, one again shown that the

proposed theoretical basis in this paper, as well as the computer program is built base on this
theory, is correct and logical. It is used to detect the maxima points (the edge points of
objects) and approximate the geological boundaries.

- *Advantage*: Zoom in and view the results on *figures 5c, 5d* and *figure 6c, 6d* and

*figure 7*, we can see that the maxima points order are detected by new approach are more
stable, more near at the real edge than Blakely's approach. Therefore, using the blue points to
approximate the edge of objects will show more clearly in both case.

- *Disadvantage:* It is selection n>=2. For model 1, we can choose n>=2 for both

approachs. But for model 2, with the new approach, if we choose n>=2, it is loose. Therefore,
the paper chose n>2 for both cases (hasn't noise and has noise). In the case has noise, it is
more tight, has more points than the Blackely's approach (*fig.7*).

**4. Real data application**

Hereafter, the paper presents the results of the application of our new approach to

detect the maxima points of the horizontal gradient amplitude function of the second vertical
derivative of the gravity field and approximate the structural boundaries by the bouguer
gravity anomaly data in the East Vietnam Sea that is directly calculated from Free-air gravity
anomaly data [27] and seabed topography [27] by Parker's algorithm [22], both data sources
have scale 1minute.

The application area has the coordinate: $108^0E$-$116^0E$, $6^0N$-$18^0N$. The bouguer gravity

anomaly on research area has the fluctuation value from -35mgal to 325mgal.

The gravity field g which is measured by gravimeter varies with height, that is, there

is vertical gradient $g_z$. Over a non-uniform earth in which density varies laterally, the vertical
changes and the rate of change $g_{zz}$ is thus the second vertical derivative of the gravity field $g_z$.

Therefore, from the bouguer gravity anomaly, the paper make the upward

continuation at other altitudes, including: 10, 15, 20, 25, 30km. These calculative steps can
remove a great part of the residual anomalies (the effect of the shortest wavelengths) that is
caused by the seabed terrain or the local geology structures [15]. Each these upward
continued gravity fields is calculated the first vertical derivative. Each fields obtained after
derivation is used to calculate the horizontal gradient and the horizontal gradient amplitude.
The maxima points are detected from these horizontal gradient amplitude fields by both
approach (choose n>=2 for both). The obtained results are shown on figures *8a, 8b,8c,8d,*
*8e,8f and 9.*

Figures (*figure 8a, 8b,8c,8d, 8e,8f*) that a part of results is zoomed in for we can view

more detailer about the difference between results of two approach at other altitudes. In





which, the yellow points are detected by our approach, the red points are detected by
Blackely's approach and the horizontal gradient vectors are shown by the cyan color (the
amplitude is multiplied with 10). We can see that many points (yellow points) are dectected
by our new approach but aren't detected by Blackely'approach at any altitudes. There, their
horizontal gradient vectors have small amplitude and quite confusional direction. These
maxima points are approximated by the green polylines (*Fig. 8f*). We believe that the green
polylines are a new boundary because it wasn't shown in the projects and articles [9, 14].
This boundary is only detected base on our algorithm by the bouguer gravity anomaly. In the
future, this boundary can be verified by the other geophysics methods, as well as other
geology results.

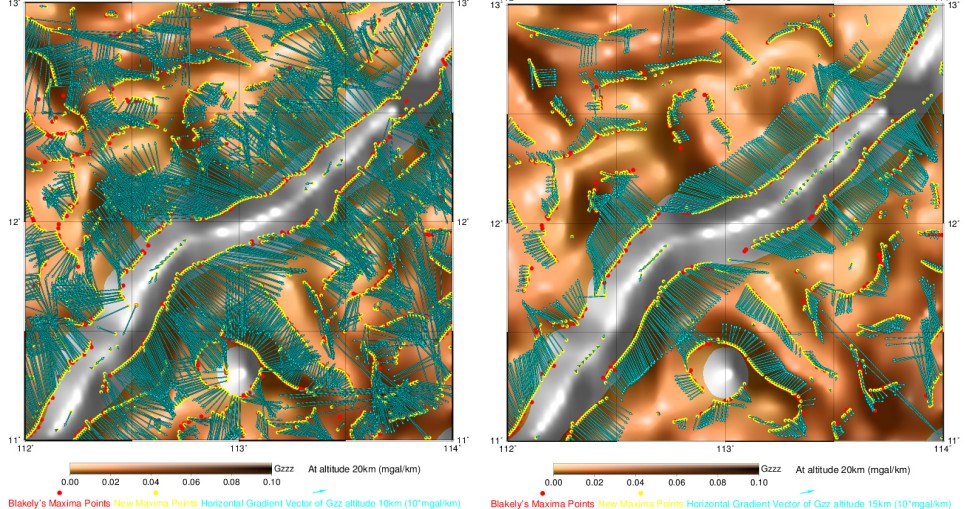


*Figure 8*a). Altitude 10 km                    *Figure 8*b). Altitude 15 km


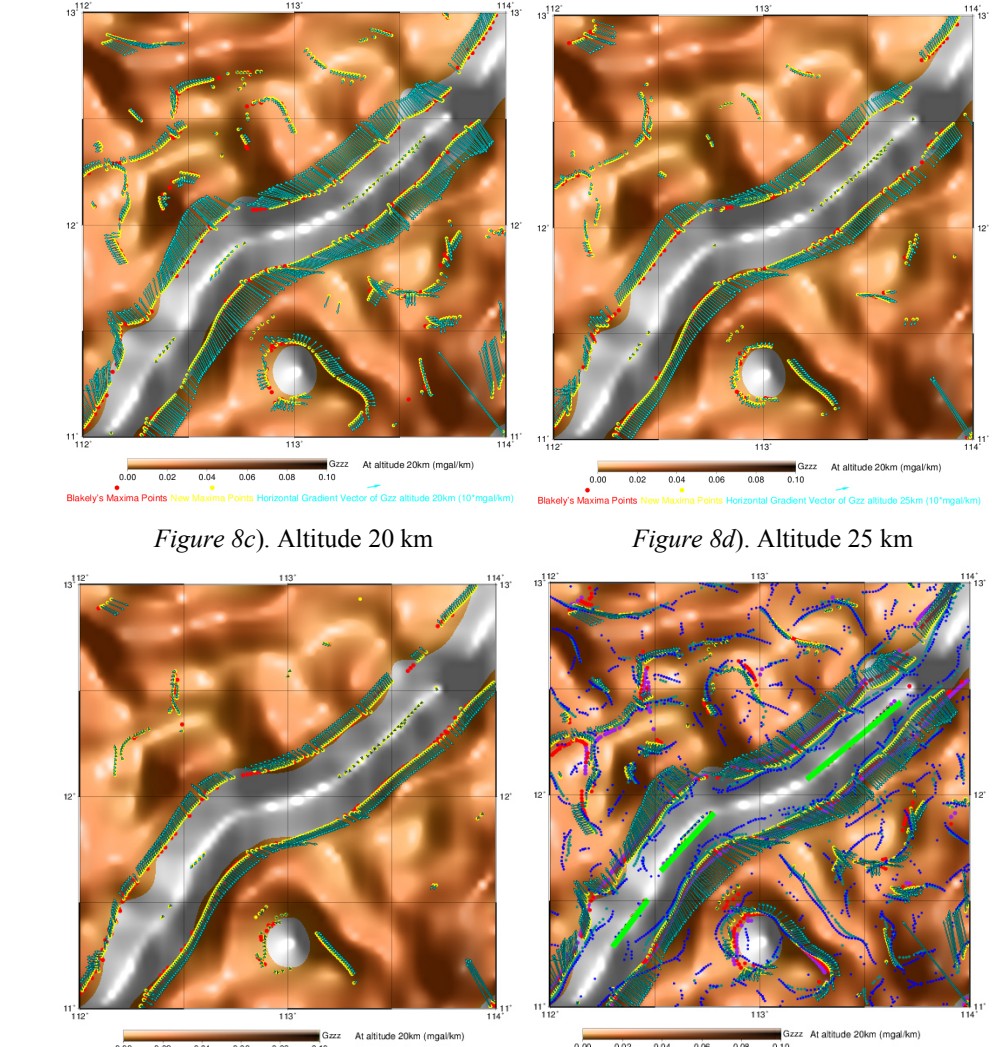


*Figure 8c)*. Altitude 20 km

*Figure 8d)*. Altitude 25 km


*Figure 8e)*: Altitude *30* km

*Figure 8f): The maxima points are detected by new approach at other altitudes (the green polylines are the approximated boundary that is only detected by this approach)*

The maxima points that are detected at other altitudes base on our new approach in

the East Vietnam Sea are shown on *figure 9*

*Figure 9.* The maxima points are detected base on new approach by the bouguer gravity

anomaly in the East Vietnam Sea

(*the green polylines are the approximated boundaries that is only detected by our approach*)





## 5. Conclusion

Originating from two-variables function that the coefficients are established by Gauss elimination method base on 9 data points and examine on their specific cases (including: case x=0, y=0 correspond with two quadratic functions, case y=-x, y=x correspond with two quartic functions) the paper proposed a the theoretical basis, a algorithm to detect the maxima points.

Base on the theoretical basis that the paper proposed, the paper built a computer program by the Matlab computer programming language to detect the extreme points, the maximum point on 4 dimensions correspond with 4 functions of one variable. From the results tested on the numberic models shown that the built computer program, as well as the proposed theoretical basis, is correct and logical.

From the results obtained on the numberic models and applied on real data shown that the maxima points are detected by new approach have more points than Blakely's approach. Therefor, Using the maxima points are detected by new approach to approximate the edge of objects are better.

We conclude with the application of our new approach to gravitational data in the East Vietnam Sea and demonstrate that we thereby disclose the existence of a gravity trench undetectable in the traditional method.



**APPENDIX A**

**Solving the specific cases of two-variables function**


***1). The specific cases of two-variables function***
$f_{(x,y)} = a_1 x^2 + a_2 y^2 + a_3 x^2 y^2 + a_4 x^2 y + a_5 xy^2 + a_6 xy + a_7 x + a_8 y + a_9;$                    (A-1)

* Case: x=0:          $f_{(x,y)}^{x=0} = a_2 y^2 + a_8 y + a_9;$                              (A-2)

* Case: y=0:          $f_{(x,y)}^{y=0} = a_1 x^2 + a_7 x + a_9;$                              (A-3)

* Case: y=-x: $f_{(x,y)}^{y=-x} = a_3 x^4 + (a_5 - a_4)x^3 + (a_1 + a_2 - a_6)x^2 + (a_7 - a_8)x + a_9;$ (A-4)
*Case: y=x:  $f_{(x,y)}^{y=x} = a_3 x^4 + (a_5 + a_4)x^3 + (a_1 + a_2 + a_6)x^2 + (a_7 + a_8)x + a_9;$ (A-5)

To detect the extreme points of these cases, we have to calculate the first-order

derivative these functions (equation A-2, A-3, A-4,A-5) and solve these:

***\* Case: x=0***: $2a_2 y + a_8 = 0$                              *(A-6)*

therefore: $y_m = \dfrac{-a_8}{2a_2}$; and replace into equation A-2,we  tain:
$g_{\max}^{x=0} = a_2 y_m^2 + a_8 y_m + a_9;$ (A-7)

if : $-dy < y_m = \dfrac{-a_8}{2a_2} < 0;$  and  $g_8 \le g_{\max}^{x=0} \ge g_5;$    (call is segment 8-5);(A-8)

if: $0 < y_m = \dfrac{-a_8}{2a_2} < dy;$  and  $g_2 \le g_{\max}^{x=0} \ge g_5;$       (segment 2-5);    (A-9)

***\* Case: y=0***: $2a_1 x + a_7 = 0$                              *(A-10)*

therefore: $x_m = \dfrac{-a_7}{2a_1}$, and replace into eq. A-3,we  obtain: $g_{\max}^{y=0} = a_1 x_m^2 + a_7 x_m + a_9;$ (A-11)

if : $-dx < x_m = \dfrac{-a_7}{2a_1} < 0;$  and  $g_4 \le g_{\max}^{y=0} \ge g_5;$    (segment 4-5);   (A-12)

if: $0 < x_m = \dfrac{-a_7}{2a_1} < dx;$  and  $g_6 \le g_{\max}^{y=0} \ge g_5;$       (segment 6-5);   (A-13)

***\* Case: y=-x***: $4a_3 x^3 + 3(a_5 - a_4)x^2 + 2(a_1 + a_2 - a_6)x + (a_7 - a_8) = 0$       *(A-14)*





With    $\begin{aligned} a &= 4a_3; \\ b &= 3(a_5 - a_4); \\ c &= 2(a_1 + a_2 - a_6); \\ d &= a_7 - a_8; \end{aligned}$
We obtain: $ax^3 + bx^2 + cx + d = 0;$    (A-15)
***Case: y=x**: $4a_3 x^3 + 3(a_5 + a_4)x^2 + 2(a_1 + a_2 + a_6)x + (a_7 + a_8) = 0;$    *(A-16)*
With    $\begin{aligned} a &= 4a_3; \\ b &= 3(a_5 + a_4); \\ c &= 2(a_1 + a_2 + a_6); \\ d &= a_7 + a_8; \end{aligned}$
We obtain: $ax^3 + bx^2 + cx + d = 0;$    (A-17)
To solve equations (A-15 and A-17) we use *Appendix below: 2). Solving cubic*
*equation*
The roots $x_m^{y=-x}$ (case y=-x) and $x_m^{y=x}$ (case y=x) are determined by expression
from A-25 to A-27 or A-28 or A-29 or A-30.
Case y=-x, we will have $y_m = -x_m$. Replace ($x_m^{y=-x}$, $y_m$) into (A-4) we have $g_{max}^{y=-x}$
Case y=x, we will have $y_m = x_m$. Replace ($x_m^{y=x}$, $y_m$) into (A-5) we have $g_{max}^{y=x}$
Now, we examine:
if: $-dx < x_m^{y=-x} < 0;$ and $g_1 \le g_{max}^{y=-x} \ge g_5;$ (call is segment 1-5);    (A-18)
if: $0 < x_m^{y=-x} < dx;$ and $g_9 \le g_{max}^{y=-x} \ge g_5;$    (segment 9-5);    (A-19)
if: $-dx < x_m^{y=x} < 0;$ and $g_7 \le g_{max}^{y=x} \ge g_5;$    (segment 7-5)    (A-20)
if: $0 < x_m^{y=x} < dx;$ and $g_3 \le g_{max}^{y=x} \ge g_5;$    (segment 3-5);    (A-21)
Like this, we have 4 directions and are separated into 8 segments. To have a
maximum point, we need choose n>= 2 (conditional satisfiable segments on total 8 segments
(A-8, A-9, A-12, A-13 and A-18 to A-21)).

***2). Solving cubic equation***
Supposing that we have a cubic equation:
$ax^3 + bx^2 + cx + d = 0;$ (with a # 0) ;    (A-22)
$\Delta = b^2 - 3ac$    (A-23)



$$k = \frac{9abc - 2b^3 - 27a^2d}{2\sqrt{|\Delta|^3}}$$
(A-24)

1). If $\Delta > 0$ and $|k| \leq 1$ , equation (A-22) has 3 roots:
$$x_1 = \frac{2\sqrt{\Delta}\cos(\frac{\arccos(k)}{3}) - b}{3a};$$
(A-25)

$$x_2 = \frac{2\sqrt{\Delta}\cos(\frac{\arccos(k)}{3} - \frac{2\pi}{3}) - b}{3a};$$
(A-26)

$$x_3 = \frac{2\sqrt{\Delta}\cos(\frac{\arccos(k)}{3} + \frac{2\pi}{3}) - b}{3a};$$
(A-27)

2). If $\Delta > 0$ and $|k| > 1$ , equation (A-22) only has one root:
$$x = \frac{\sqrt{\Delta}|k|}{3ak}(\sqrt[3]{|k| + \sqrt{k^2 - 1}} + \sqrt[3]{|k| - \sqrt{k^2 - 1}}) - \frac{b}{3a}$$
(A-28)

3). If $\Delta = 0$, equation (A-22) has multiples roots:
$$x = \frac{-b + \sqrt[3]{b^3 - 27a^2d}}{3a};$$
(A-29)

4). If $\Delta < 0$, equation (A-22) only has one root
$$x = \frac{\sqrt{|\Delta|}}{3a}(\sqrt[3]{k + \sqrt{k^2 + 1}} + \sqrt[3]{k - \sqrt{k^2 + 1}}) - \frac{b}{3a}$$
(A-30)

**Acknowledgements**
**The author sincerely thank for the projects of VAST (code: QTRU02.01/19-20,**
**QTRU02.01/20-21, VAST06.01/20-21) and KHCBTĐ.02/18-20 project, VT-UD.04/17-20**
**project supported conditions for complete this paper.**



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
