# Peer review of "A Gauss Elimination Method for estimating locations of extrema in gridded data: Applications for Potential Field Data"

_Nonlinear Processes in Geophysics, 2020_

## Referee Comment (RC1) · Anonymous Referee #1 · 28 Aug 2020

Discussion: Overall, the language is poor and is an obstacle to understand the manuscript, and to showcase your algorithm. I am personally struggling with your algorithm. Why using Gauss elimination on a function of the type Eq. 1 where linear programming and optimization methods are nowadays much more powerful. You need to do a discussion on this point. In terms of results, I think the "Test Cases" are poorly written, and again difficult to follow. The level of noise is a crucial parameter in edge detection. Therefore, you must show differently the performances of your algorithm. The "Real Data" is interesting, but not much is said on the dataset. I am also left with many questions on the conclusions as you wrote some bold statements such as "we believe line 231-232", which is not supported enough. In the results part, I believe that

a statistical test is missing on the edge detection to compare your results with other methods. In summary, the work is interesting, but the algorithm is not showcased with the current state of the manuscript. Therefore, I can only reject it, but I encourage you to resubmit a new manuscript. Below, I have made some comments to help you to improve the manuscript.

Major comments

- Introduction: After reading the introduction, I cannot say what is new in this manuscript. Thus, the introduction should be revised to lead the reader. You could for example makes a Table listing the weak points and strong points of all the methods cited in the introduction. 2/ be precise when using statements such as l.58 ". . . function that is established by three points on a straight line. or l.58 "the accuracy of the approximated geological boundaries aren't high enough", or "..these functions are very different from the functions that the proposed" – avoid bold statements!

- Introduction: you need to define "Gauss elimination method". That is a generic mathematical definition which has applications in many scientific areas.

- Method: l.76 "paper researches a function of two variables that has pattern" – why? I am not an expert in this field and there is nothing to introduce this type of function. Is it from previous work? Literature?

- Method: Table 1 is interesting. I like it. Thus, I would take this table and put it in the introduction, expanding it with other methods. It can also give a nice introduction to your method.

- Method: l. 103 "However, this paper doesn't detect the critical points . . ." I am lost. When I read the manuscript I am confused, because 1/ you estimate the critical point at fx =0, fy=0, why? 2/ l.104 you now speak about extreme points. You need to rewrite the section and show your assumptions, define critical and extreme points, how you detect them or why estimating at specific point in space. The assumptions are crucial

to see if the software will not be biased.

- Method: when I see that you want to solve Eq. (1), you can use much more advanced mathematical tools such as optimization algorithm (gradient descent, second order algorithm > Jacobi-Davidson), convex optimization . . .. Thus, you should justify why using your method, and above all why your algorithm will not be biased due to the basic assumptions.

- Test Cases: I would prefer to rename this section, with perhaps "Results and Discussion" – you should also divide in two parts – "simulations" and "real data"

- Test Cases: you should expend that you know use Matlab (with proper reference) to use your algorithm on simulated data set.

- Test Cases: Table 2 shows the parameters of the models used to do the simulations, but we do not see this model – I believe it is mentioned in line 120 "Mantik Talwni and Maurice Ewing [26]". Thus, you need to give the model at least in the appendices.

- Test Cases: I think the whole presentation of the results should be revised. The "model hasn't noise" or "the model has noise" should be fused together with proper words and explain the limits of the algorithm when using higher noise level. Perhaps, a figure showing the statistics testing for edge detection of the figures in various noise levels could be much more understandable. Also, the use of "comments " (l.161) or "advantage/disadvantage" (l194,198) does break the flow of the paper. Again, proper words and text management can improve a lot the manuscript.

- Real data: You need to properly introduce the data set.

- Real data: to support your conclusions about your algorithm more sensitive to the relief of the image, you need to look in other packages about edge detections and compared with yours. The type of sentence l. 231 " We believe that the green 231 polylines are a new boundary because it wasn't shown in the projects " are not supported – thus we can wonder if there is noise included in the edge detection. . . or else

. . . a full discussion is required here.

Minor mistakes

Please revise the English - here are a few mistakes -

l. 36 revise the sentence "We have many the methods . . ." l.49 revise "Each author, as well as each method, has . . ." l.50 "detect" > to detect l.51 Revise and simplify the two sentences "These methods are very powerful. They confirmed on many papers and projects of authors, . . . " I have never cited a method or an algorithm by the name of the author, but by references. It is also confusing to add the name and the reference. Thus, lines 47-49 and lines 55-56 should be revised. Avoid double referencing name + reference. Perhaps, you can simply say "method developed by [24] will be referred as the method of Philips in the following " l.59 avoid using the verbal contraction "aren't" in any scientific manuscripts, always write the full verbal structure. Also same for "isn't" (l.61, 65, . . . ) and hasn't (l.127 ..), wasn't (l.231) l.60 Function > function l.64 "," before "because" l.69 "the marked differences" – what does that mean? l.113 Revise : "the paper use the built computer program to test on two numerical model".

---

## Referee Comment (RC2) · Anonymous Referee #2 · 8 Sep 2020

Paper discusses gravity measurements and minima estimation by using a simple polynomial approximation and Gaussian elimination. The application is interesting from the NPG point of view, however, the methodology presented is trivial, and as such there is no new contribution to the scientific literature in the sense of methodology development. Language, structure, and the equations presented are not up to the standard of international peer-reviewed publication.

I propose rejection due to incrementality and substandard writing.

---

## Short Comment (SC1) · 8 Sep 2020

**First of all, thank you for your comments and suggestions that allowed us to greatly improve the quality of the manuscript**

**1. Discussion**

**1.1. Why using Gauss elimination on a function of the type Eq. 1 ?**

*Answer:* In the introduction, we discussed about disadvantage and advantage of two methods:the method of Blackely, R. J., Simpson, R.W, 1986 and the method of Phillips, J.D. 2007. This paper is a combination of these methods. Gauss elimination method is used to determine the coefficients of two-variables function (eq.1).

**1.2. Why using eq.1 ?**

*Answer:* Eq.1 has type general $(x-a)^{\wedge 2}(x-b)^{\wedge 2}$.

- Due to 3x3 data grid has 9 point. It is 9 equations. We has matrix $\mathbf{Ax=b}$. Inthere, $\mathbf{A}$ are co-ordinate (x,y) of data points, $\mathbf{x}$ are coefficients of Eq1, $\mathbf{b}$ is data

- Gauss elimination method to solve for these equations has unique root.

**1.3. Why combining these methods ?**

*Answer:*Using Gauss elimination method to determine the coefficients of two-variables function. Afterthat, we only examine on 4 specical cases of this function, include: x=0, y=0, y=-x, y=x. They are 4 cases that Blackely introduced.

- In general, the Gauss elimination method used for stage 1 (establish a two-variables function base on 3x3 data grid) and the method of Blackely for stage 2 (calculate the maxima from 4 functions).

**2. Major comments:**

**2.1. Comment 1:**

- Introduction:…………. l.58 "… function that is established by three points on a straight line or l.58 "the accuracy of the approximated geological boundaries aren't high enough", or "..these functions are very different from the functions that the proposed". – avoid bold statements.

*Answer:* They will be revised with more proper words.

**2.2. Comment 2:**

- Introduction: you need to define " Gauss elimination method"….

*Answer:* Gauss elimination method used popular therefor we don't put into this paper. It can be referenced at: https://en.wikipedia.org/wiki/Gaussian_elimination

**2.3. Comment 3:**

- Method: l.76 "paper researches a function of two variables that has pattern" why? I am not an expert in this field and there is nothing to introduce this type of function. Is it from previous work? Literature?

*Answer:* No, this paper dosen't researches about function of two variables. This paper only use a type of two-variables function. I think it dosen't need to introduce in this paper. It can be found in the paper of Phillips, J.D. 2007 [24].

**2.4. Comment 4:**

- Method: Table 1 is interesting. I like it. Thus, I would take this table and put it in the introduction, expanding it with other methods. It can also give a nice introduction to your method.

*Answer:* Thank you for your comments, but I think this table should put there because it easy to view and discuss

**2.5. Comment 5:**

- Method: l. 103 "However, this paper doesn't detect the critical points ..." I am lost. When I read the manuscript I am confused, because 1/ you estimate the critical point at fx =0, fy=0, why? 2/ l.104 you now speak about extreme points. You need to rewrite the section and show your assumptions, define critical and extreme points, how you detect them or why estimating at specific point in space. The assumptions are crucial *to* see if the software will not be biased.

*Answer:* This paper doesn't detect the critical points from function $f_{(x,y)}$ (two-variable function). This paper detect the extreme points from one-variable function (is specific cases of $f_{(x,y)}$)

For this paper: A critical point may be: a local maximum, local minimum, or saddle point for a function of two variable. A extreme point may be: a local maximum or local minimum for a function of one variable

**2.6. Comment 6:**

- Method: when I see that you want to solve Eq. (1), you can use much more advanced mathematical tools such as optimization algorithm (gradient descent, second order algorithm > Jacobi-Davidson), convex optimization ... Thus, you

should justify why using *your* method, and above all why your algorithm will not be biased due to the basic assumptions.

*Answer:* Each method has advantage and disadvantage, we use Gauss elimination method because this method for unique root. Therefor, it is a new approach

**2.7. Comment 7.**

- Test Cases: I would prefer to rename this section, with perhaps "Results and Discussion" – you should also divide in two parts – "simulations" and "real data

*Answer:* Thank you for your comments, may be we will rename them

**2.8. Comment 8.**

- Test Cases: you should expend that you know use Matlab (with proper reference) to use your algorithm on simulated data set.

*Answer:* Thank you for your comments, we will expend for next paper.

**2.9. Comment 9.**

- Test Cases: Table 2 shows the parameters of the models used to do the simulations,but we do not see this model – I believe it is mentioned in line 120 "Mantik Talwni and Maurice Ewing [26]". Thus, you need to give the model at least in the appendices

*Answer:* Table 2 has the parameters of both two models (model 1 and model 2) include: local, depth (top and bottom), desity constrat

On Fig 2 show local and gravity anomaly of objects for both two models

Line 120 "Mantik Talwni and Maurice Ewing [26]" is a Matlab code that is built base on the theoretical basis of Mantik Talwni and Maurice Ewing to calculate gravity anomaly of objects in model 2

**2.10. Comment 10.**

- Test Cases: I think the whole presentation of the results should be revised. The "model hasn't noise" or "the model has noise" should be fused together with proper words and explain the limits of the algorithm when using higher noise level. Perhaps,a figure showing the statistics testing for edge detection of the figures in various noise levels could be much more understandable. Also, the use of "comments " (L.161) or "advantage/disadvantage" (L194,198) does break the flow of the paper. Again, proper words and text management can improve a lot the manuscript.

*Answer:* Thank you for your comments, this comment is very good, but we want to introduce the models with more difficult level

**2.11. Comment 11**

- Real data: You need to properly introduce the data set.

*Answer:* The data set is introduced at l.207, 208.

**2.12. Comment 12.**

- Real data: to support your conclusions about your algorithm more sensitive to the relief of the image, you need to look in other packages about edge detections and compared with yours. The type of sentence l. 231 " We believe that the green 231 polylines are a new boundary because it wasn't shown in the projects " are not supported – thus we can wonder if there is noise included in the edge detection:.. or else ... a full discussion is required here.

*Answer:* The paper want to introduce a new approach to dectect the edge of objects **base on potential field**. We haven't more information of other packages such as seismic,….It is very difficult and expensive to have a seismic cross section at this area. Our projects aren't enough money. Therefor, we only can base on the results of projects previous to compare. We hope that this new boundary may be detected and discoved by other methods from the bigger projects.

**3. Minor mistakes**

Please revise the English - here are a few mistakes

L.36 revise the sentence "We have many the methods ...". l.49 revise "Each author, as well as each method, has ...".  l.50 "detect" > to detect. l.51 Revise and simplify the two sentences "These methods are very powerful. They confirmed on many papers and projects of authors,..." I have never cited a method or an algorithm by the name of the author, but by references. It is also confusing to add the name and the reference. Thus, lines 47-49 and lines 55-56 should be revised. Avoid double referencing name + reference. Perhaps, you can simply say "method developed by [24] will be referred as the method of Philips in the following " l.59 avoid using the verbal contraction "aren't" in any scientific manuscripts, always write the full verbal structure. Also same for "isn't" (l.61, 65, .... ) and hasn't (l.127 ..), wasn't (l.231) l.60 Function > functionl.64 "," before "because" l.69 "the marked differences" – what does that mean? l.113 Revise : "the paper use the built computer program to test on two numerical model".

*Answer:* Thank you for your comments, the English mistakes will be revised

---

## Author Comment (AC1) · 28 Feb 2021

**First of all, thank you for your comments and suggestions that allowed us to greatly improve the quality of the manuscript**

**1. Discussion**

**1.1. Why using Gauss elimination on a function of the type Eq. 1 ?**

*Answer:* In the introduction, we discussed about disadvantage and advantage of two methods:the method of Blackely, R. J., Simpson, R.W, 1986 and the method of Phillips, J.D. 2007. This paper is a combination of these methods. Gauss elimination method is used to determine the coefficients of two-variables function (eq.1).

**1.2. Why using eq.1 ?**

*Answer:* Eq.1 has type general $(x-a)^{\wedge 2}(x-b)^{\wedge 2}$.

- Due to 3x3 data grid has 9 point. It is 9 equations. We has matrix **Ax=b**. Inthere, **A** are co-ordinate (x,y) of data points, **x** are coefficients of Eq1, **b** is data

- Gauss elimination method to solve for these equations that has unique root.

**1.3. Why combining these methods ?**

*Answer:*Using Gauss elimination method to determine the coefficients of two-variables function. Afterthat, we only examine on 4 specical cases of this function, include: x=0, y=0, y=-x, y=x. They are 4 cases that Blackely introduced.

- In general, the Gauss elimination method use for step 1 (establish a two-variables function base on 3x3 data grid). The method of Blackely use for step 2 (calculate the maxima from 4 functions).

**2. Major comments:**

**2.1. Comment 1:**

- Introduction:…………. l.58 "… function that is established by three points on a straight line or l.58 "the accuracy of the approximated geological boundaries aren't high enough", or "..these functions are very different from the functions that the proposed". – avoid bold statements.

*Answer:* The paper is revised.

**2.2. Comment 2:**

- Introduction: you need to define " Gauss elimination method"….

*Answer:* Gauss elimination method used popular. It can be referenced at: https://en.wikipedia.org/wiki/Gaussian_elimination

**2.3. Comment 3:**

- Method: l.76 "paper researches a function of two variables that has pattern" why? I am not an expert in this field and there is nothing to introduce this type of function. Is it from previous work? Literature?

*Answer:* No, this paper dosen't researches about function of two variables. This paper only use a type of two-variables function. Function of two variables can be found in the paper of Phillips, J.D. 2007 .

**2.4. Comment 4:**

- Method: Table 1 is interesting. I like it. Thus, I would take this table and put it in the introduction, expanding it with other methods. It can also give a nice introduction to your method.

*Answer:* Thank you for good comments, we will review

**2.5. Comment 5:**

- Method: l. 103 "However, this paper doesn't detect the critical points ..." I am lost. When I read the manuscript I am confused, because 1/ you estimate the critical point at fx =0, fy=0, why? 2/ l.104 you now speak about extreme points. You need to rewrite the section and show your assumptions, define critical and extreme points, how you detect them or why estimating at specific point in space. The assumptions are crucial *to* see if the software will not be biased.

*Answer:* Thank you for comments, The paper is revised

**2.6. Comment 6:**

- Method: when I see that you want to solve Eq. (1), you can use much more advanced mathematical tools such as optimization algorithm (gradient descent, second order algorithm > Jacobi-Davidson), convex optimization ... Thus, you should justify why using *your* method, and above all why your algorithm will not be biased due to the basic assumptions.

*Answer:* Each method has advantage and disadvantage, we use Gauss elimination method because this method for unique root. Therefor, it is a new approach

**2.7. Comment 7.**

- Test Cases: I would prefer to rename this section, with perhaps "Results and Discussion" – you should also divide in two parts – "simulations" and "real data

*Answer:* Thank you for good comments, we will review

**2.8. Comment 8.**

- Test Cases: you should expend that you know use Matlab (with proper reference) to use your algorithm on simulated data set.

*Answer:* Thank you for comments, we will review.

**2.9. Comment 9.**

- Test Cases: Table 2 shows the parameters of the models used to do the simulations,but we do not see this model – I believe it is mentioned in line 120 "Mantik Talwni and Maurice Ewing [26]". Thus, you need to give the model at least in the appendices

*Answer:* Table 2 has the parameters of both two models (model 1 and model 2) include: local, depth (top and bottom), desity constrat

On Fig 2 show local and gravity anomaly of objects for both two models

Line 120 "Mantik Talwni and Maurice Ewing [26]". A Matlab code that is built base on the theoretical basis of  Mantik Talwni and Maurice Ewing to calculate gravity anomaly of objects in model 2

**2.10. Comment 10.**

- Test Cases: I think the whole presentation of the results should be revised. The "model hasn't noise" or "the model has noise" should be fused together with proper words and explain the limits of the algorithm when using higher noise level. Perhaps,a figure showing the statistics testing for edge detection of the figures in various noise levels could be much more understandable. Also, the use of "comments " (L.161) or "advantage/disadvantage" (L194,198) does break the flow of the paper. Again, proper words and text management can improve a lot the manuscript.

*Answer:* Thank you for comments. The paper is revised.

**2.11. Comment 11**

- Real data: You need to properly introduce the data set.

*Answer:* The data set is introduced.

**2.12. Comment 12.**

- Real data: to support your conclusions about your algorithm more sensitive to the relief of the image, you need to look in other packages about edge detections and compared with yours. The type of sentence l. 231 " We believe that the green 231 polylines are a new boundary because it wasn't shown in the projects " are not supported – thus we can wonder if there is noise included in the edge detection:.. or else ... a full discussion is required here.

*Answer:* The paper introduce a new approach to dectect the edge of objects base on potential field.

**3. Minor mistakes**

Please revise the English - here are a few mistakes

L.36 revise the sentence "We have many the methods ...". l.49 revise "Each author, as well as each method, has ...".  l.50 "detect" > to detect. l.51 Revise and simplify the two sentences "These methods are very powerful. They confirmed on many papers and projects of authors,..." I have never cited a method or an algorithm by the name of the author, but by references. It is also confusing to add the name and the reference. Thus, lines 47-49 and lines 55-56 should be revised. Avoid double referencing name + reference. Perhaps, you can simply say "method developed by [24] will be referred as the method of Philips in the following " l.59 avoid using the verbal contraction "aren't" in any scientific manuscripts, always write the full verbal structure. Also same for "isn't" (l.61, 65, .... ) and hasn't (l.127 ..), wasn't (l.231) l.60 Function > functionl.64 "," before "because" l.69 "the marked differences" – what does that mean? l.113 Revise : "the paper use the built computer program to test on two numerical model".

*Answer:* Thank you for good comments, the English mistakes is revised

---

## Author Comment (AC2) · 28 Feb 2021

**First of all, thank you for your comments and suggestions that allowed us to greatly improve the quality of the manuscript**

**1. Discussion**

Paper discusses gravity measurements and minima estimation by using a simple poly-nomial approximation and Gaussian elimination. The application is interesting from theNPG point of view, however, the methodology presented is trivial, and as such thereis no new contribution to the scientific literature in the sense of methodology develop-ment. Language, structure, and the equations presented are not up to the standard ofinternational peer-reviewed publication.

I propose rejection due to incrementality and substandard writing

*Answer:* The paper is revised.